# Association of high-sensitivity C-reactive protein to albumin ratio with all-cause and cardiac death in coronary heart disease individuals: A retrospective NHANES study

Shangxun Zhou[1,2], Miaohan Qiu[1], Kexin Wang[1], Yixuan Duan[1,2], Daoshen Liu[1,2], Ying Xu[1], Xuefei Mu[1], Jing Li[1], Yi Li[1], Yaling Han[1]*

1 State Key Laboratory of Frigid Zone Cardiovascular Disease, Cardiovascular Research Institute and Department of Cardiology, General Hospital of Northern Theater Command, Shenyang, China, 2 The Department of Cardiology, Xijing Hospital, Air Force Medical University, Xi'an, China

* hanyaling@163.net

## Abstract

### Background

This research aimed to explore the association of high-sensitivity C-reactive protein to albumin ratio (CAR) with death events in community-based patients with coronary heart disease (CHD).

### Methods

624 CHD participants were followed for 36 months using data from the 2015–2018 National Health and Nutrition Examination Survey (NHANES). The CAR was dichotomized at 0.075 mg/g to stratify inflammation levels. Relationships between CAR, high-sensitivity C-reactive protein (hsCRP), albumin (ALB) and all-cause and cardiac death in all participants and subgroups were analyzed using restricted cubic spline (RCS), Kaplan-Meier survival curves and Cox proportional hazards models.

### Results

Both CAR and hsCRP showed positive correlations with all-cause and cardiac death risk while ALB exhibited a U-shaped correlation with all-cause death risk but a negative correlation with cardiac death risk. The high-CAR group had higher risks of all-cause (P = 0.04) and cardiac death (P = 0.02). The hazard ratios (HR) (95% confidence intervals (CI)) for all-cause death was 1.77 (1.15–2.74) (P = 0.010), while it was 2.99 (1.44–6.22) (P = 0.003) for cardiac death. No significant interaction was observed in subgroup analyses.

**Data availability statement:** All data files are available from the NHANES database (https://www.cdc.gov/nchs/nhanes/).

**Funding:** The study was supported by the National Key Research and Development Program of China (2022YFC2503500; 2022YFC2503504). The funders had no role in study design, data collection and analysis, decision to publish, or preparation of the manuscript.

**Competing interests:** The authors have declared that no competing interests exist.

## Conclusions

A CAR threshold of 0.075 mg/g effectively distinguished between high and low inflammation risks. Elevated CAR significantly increased the risk of all-cause and cardiac death in community CHD patients.

## Introduction

Cardiovascular disease (CVD) is a leading cause of global mortality, highlighting the critical importance of early risk assessment and preventive measures to improve patient outcomes [1–5]. Investigating the correlation between various biomarkers and CVD is crucial for identifying high-risk individuals who require timely intervention. Previous research has emphasized the significant impact of acute and chronic inflammatory responses on CVD development, affecting important cellular processes such as autophagy, apoptosis and cardiomyocyte remodeling [6–8]. High-sensitivity C-reactive protein (hsCRP) and albumin (ALB) are widely recognized as key indicators of inflammation levels [9–13]. Elevated hsCRP levels and reduced ALB levels have been linked to increased risk of CVD [14,15]. The high-sensitivity C-reactive protein to albumin ratio (CAR) or C-reactive protein to albumin has emerged as a valuable prognostic indicator in various medical conditions including solid tumors, hematological malignancies, cardiovascular diseases and respiratory diseases [16–20]. Studies have shown that elevated CAR levels are associated with higher risk of CVD while demonstrating better predictive performance compared to hsCRP or ALB alone [21]. However, existing studies on CAR have mainly categorized it into binary, ternary or quadrivalent variables based on median, triquartile or quartile values without establishing a definitive threshold for widespread use. Furthermore, research on CAR in relation to cardiovascular disease has primarily focused on critically ill patients, those with acute coronary syndrome (ACS), or community populations without prior history of CVD, there is a lack of investigation targeting community-dwelling patients with coronary heart disease (CHD). Therefore, this study aims to establish a clear and universally applicable threshold value for CAR while examining its association with all-cause and cardiac death among community-dwelling CHD patients.

## Materials and methods

### Study participants

The National Health and Nutrition Examination Survey (NHANES) was approved by the Ethics Review Board of the National Center for Health Statistics (NCHS) of the United States. It was designed to gather information on the health and nutrition of American families, with all participants providing written informed consent. The identity information of all participants was kept confidential, and they were identified solely by the respondent sequence numbers (SEQN). A total of 19,363 individuals from the database spanning 2015–2018 were identified, out of which 1189 had self-reported doctor-diagnosed CHD and symptoms of primary and secondary angina. After excluding 113 individuals with missing hsCRP or ALB data, a study cohort consisting

of 624 individuals with follow-up time ≤ 36 months was selected (Fig 1). The outcome events were defined as all-cause death and cardiac death, determined by the National Death Index (NDI) record as of December 31, 2019. The causes of all-cause death encompass diseases of the heart, malignant neoplasms, chronic lower respiratory diseases, accidents, cerebrovascular diseases, Alzheimer's disease, diabetes mellitus, influenza and pneumonia, nephropathy and other causes. All the data used in this study is publicly available (https://www.cdc.gov/nchs/nhanes/).

## Definitions and binary classification

HsCRP > 3 mg/L was considered indicative of high inflammation risk in cardiovascular disease, hence it was set as a binary classification variable using a cutoff value of 3 mg/L. $CAR = hsCRP / ALB$, where hsCRP > 3 mg/L indicated high inflammation risk and ALB < 40 g/L indicated low albuminemia. Therefore, CAR was set as a binary classification variable using a cutoff value of 0.075 mg/g to distinguish between high inflammation and low inflammation risk groups.

## Assessment of covariates

Various covariates that might influence outcomes were included in univariate and multivariate Cox regression analyses based on their association with clinical outcomes or showing P < 0.2 in univariate analysis results. The final covariates included age, gender, body mass index (BMI), diabetes, asthma, depression, segmented neutrophils, platelet (PLT), total cholesterol (TC), AST, eGFR, hypertension, chronic obstructive pulmonary disease (COPD) and cancer. Definitions are provided in the supplementary methods in S1 File.

## Statistical analysis

Data were analyzed using R 4.3.0 and SPSS 26.0. Continuous variables were expressed as mean (SD) and categorical variables as frequency (percentage). Data were compared using chi-square test and two independent sample t-test. Restricted cubic splines (RCS) were fitted to evaluate the dose-response relationship between CAR, hsCRP, ALB and the outcome events, and the nonlinearity was tested using likelihood ratio test with a threshold of P < 0.05. Kaplan-Meier cumulative incidence curves were generated to evaluate the relationship between these three variables and the risk of all-cause and cardiac death during follow-up, with group comparisons made using the log-rank test. A Cox regression model with multivariable adjustment analyzed the relationships between covariates and death events, and the hazard ratios (HR)

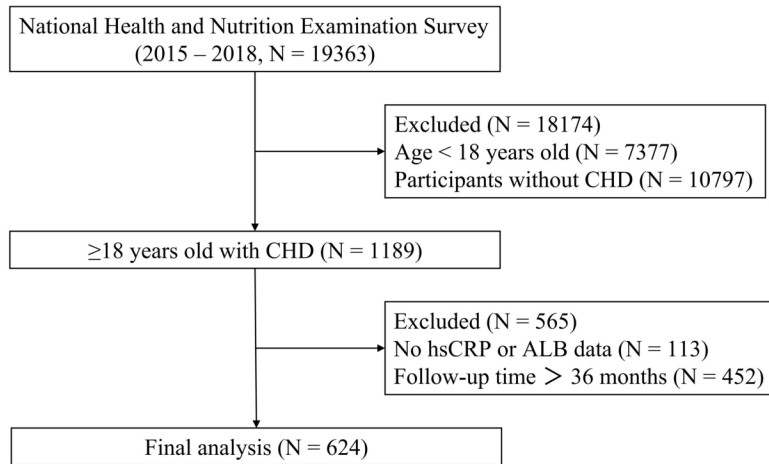

**Fig 1. The flowchart detailing the process of screening the population.**

and 95% confidence intervals (CI) were estimated. The proportional hazards assumption was tested using the Schoenfeld residual method. For subgroup analysis, the fully adjusted models were stratified by gender, age, eGFR, diabetes and COPD, and interactions were assessed. The significance level used throughout this study is P < 0.05.

## Results

### Baseline characteristics of study participants

A total of 624 patients with community-based CHD were enrolled in this study. The baseline characteristics according to the binary classification of CAR, hsCRP and ALB are presented in Table 1, S1 and S2 Tables. Among them, 41.2% were

**Table 1. Baseline characteristics of cardiovascular disease patients according to different CAR groups.**

| Variables | Total (624) | Low CAR (344) | High CAR (280) | P value |
|---|---|---|---|---|
| Age, years | 65.96 (11.77) | 67.53 (11.25) | 64.02 (12.12) | < 0.001 |
| Female, n (%) | 257 (41.2) | 121 (35.2) | 136 (48.6) | 0.001 |
| Body mass index (kg/m$^2$) | 30.24 (6.88) | 28.85 (5.66) | 31.96 (7.81) | < 0.001 |
| Physical exercise, n (%) | 298 (47.8) | 180 (52.3) | 118 (42.1) | 0.011 |
| Smoking, n (%) | | | | 0.217 |
| Never | 311 (49.8) | 179 (52.0) | 132 (47.1) | |
| Now | 95 (15.2) | 45 (13.1) | 50 (17.9) | |
| Former | 218 (34.9) | 120 (34.9) | 98 (35.0) | |
| Hypertension, n (%) | 133 (21.3) | 71 (20.6) | 62 (22.1) | 0.648 |
| Diabetes, n (%) | 234 (37.5) | 117 (34.0) | 117 (41.8) | 0.046 |
| Hyperlipidemia, n (%) | 516 (82.7) | 282 (82.0) | 234 (83.6) | 0.600 |
| Anemia, n (%) | 287 (46.0) | 132 (38.4) | 155 (55.4) | < 0.001 |
| COPD, n (%) | 156 (25.0) | 77 (22.4) | 79 (28.2) | 0.094 |
| Asthma, n (%) | 79 (12.7) | 34 (9.9) | 45 (16.1) | 0.021 |
| Depression, n (%) | 180 (28.8) | 79 (23.0) | 101 (36.1) | < 0.001 |
| Cancer, n (%) | 120 (24.2) | 64 (23.3) | 56 (25.5) | 0.574 |
| Segmented neutrophils (10$^9$/L) | 4.51 (1.82) | 4.19 (1.58) | 4.90 (2.02) | < 0.001 |
| Platelet (10$^9$/L) | 219.48 (62.88) | 209.13 (56.53) | 232.19 (67.86) | < 0.001 |
| Plasma glucose (mmol/L) | 6.40 (2.55) | 6.06 (1.78) | 6.81 (3.20) | 0.001 |
| Total cholesterol (mmol/L) | 4.51 (1.14) | 4.42 (1.07) | 4.62 (1.20) | 0.036 |
| Triglycerides (mmol/L) | 1.74 (0.98) | 1.66 (0.95) | 1.83 (1.02) | 0.031 |
| LDL-C (mmol/L) | 2.48 (0.99) | 2.36 (0.88) | 2.63 (1.11) | 0.036 |
| HDL-C (mmol/L) | 1.29 (0.37) | 1.34 (0.38) | 1.22 (0.36) | 0.001 |
| HsCRP (mg/L) | 5.36 (9.90) | 1.43 (0.76) | 10.19 (13.25) | < 0.001 |
| ALB (g/L) | 39.54 (3.56) | 40.71 (3.15) | 38.11 (3.51) | < 0.001 |
| AST (U/L) | 22.25 (10.01) | 22.73 (10.49) | 21.65 (9.38) | 0.178 |
| Creatinine (μmol/L) | 92.72 (46.78) | 90.34 (33.11) | 95.67 (26.62) | 0.177 |
| eGFR (mL/min/1.73m$^2$) | 78.13 (28.71) | 78.83 (26.62) | 77.27 (31.11) | 0.502 |
| Outcomes | | | | |
| All-cause death, n (%) | 97 (15.5) | 43 (12.5) | 54 (19.3) | 0.020 |
| Cardiac death, n (%) | 35 (5.6) | 12 (3.5) | 23 (8.2) | 0.011 |
| Cancer death, n (%) | 22 (3.5) | 10 (2.9) | 12 (4.3) | 0.353 |

Data are expressed as mean ± standard error or frequency (percentage). Abbreviations: high-sensitivity C-reactive protein to albumin ratio, CAR; chronic obstructive pulmonary disease, COPD; low-density lipoprotein cholesterol, LDL-C; high-density lipoprotein cholesterol, HDL-C; high-sensitivity C-reactive protein, hsCRP; albumin, ALB; Aspartate Aminotransferase, AST; estimated glomerular filtration rate, eGFR.

females, with an average age of 65.96 ± 11.77 years, an average hsCRP of 5.36 ± 9.90 mg/L, and an average albumin of 39.54 ± 3.56 g/L. During a median follow-up period of 23 months, the all-cause death rate was 15.5%, the cardiac death rate was 5.6%, and the cancer death rate was 3.5%. Participants with higher levels of hsCRP and CAR were more likely to be females and current smokers. They also had higher BMI, platelet counts, plasma glucose, total cholesterol, triglyceride and low-density lipoprotein cholesterol, lower high-density lipoprotein cholesterol and albumin, and are prone to diabetes, anemia, COPD and asthma. Participants with lower ALB were more likely to be females. They also had higher BMI, plasma glucose, hsCRP and creatinine, lower eGFR, and are prone to diabetes and anemia. All differences were statistically significant (P < 0.05).

### Quantitative-response relationships of CAR, hsCRP, and ALB with mortality analyzed by RCS

The RCS curves (Fig 2) showed that CAR and hsCRP were positively and linearly associated with the risk of all-cause and cardiac death. Each one-unit increase in CAR is associated with a 78% increase in the risk of all-cause death (HR: 1.78, 95%CI: 1.17–2.69, p = 0.007) and a 164% increase in the risk of cardiac death (HR: 2.64, 95%CI: 1.45–4.80, p = 0.001). The trend in risk of death with increasing CAR and hsCRP was more pronounced in the lower ranges of CAR and hsCRP. ALB was U-shaped associated with all-cause death and linearly negatively associated with cardiac death. In the U-shaped association between ALB and all-cause death risk, we obtained that the inflection point was 40.88, and subsequently calculated the effect size and confidence interval on the left and right sides of the inflection point by the segmented COX regression model. On the left side of the inflection point, each 1-unit increase in ALB was associated with a 22.6% reduction in the risk of unfavorable outcomes (HR: 0.85, 95%CI: 0.79–0.92, p < 0.001). On the right side of the inflection point, the effect size (HR) was 0.98 (95%CI: 0.85 to 1.15, p = 0.837).

### Kaplan-Meier survival curves and Cox regression analysis of risk of all-cause and cardiac death

CAR, hsCRP, and ALB were categorized as described in the methodology. The Kaplan-Meier survival curve (Fig 3) demonstrated a significantly decreased survival rate of all-cause death in the high CAR group (log-rank test, P = 0.04) and the high hsCRP group (log-rank test, P = 0.02), as well as a significantly decreased survival rate of cardiac death in the high CAR group (log-rank test, P = 0.02), the high hsCRP group (log-rank test, P = 0.03) and the low ALB group (log-rank test, P = 0.01). Covariates were included in the multivariate adjusted Cox regression model, which revealed that the HR (95%CI) for all-cause death in the high CAR group was 1.77 (1.15–2.74) (P = 0.010), and the HR (95%CI) for cardiac death was 2.99 (1.44–6.22) (P = 0.003). Furthermore, the HR (95%CI) for all-cause death in the high hsCRP group was 1.88 (1.22–2.90) (P = 0.004), and the HR (95%CI) for cardiac death was 2.63 (1.28–5.43) (P = 0.009). Additionally, the HR (95%CI) for cardiac death in the high ALB group was 0.41 (0.20–0.87) (P = 0.021) (Table 2, S3 Table). If the follow-up time was extended to 60 months, a significant decrease in the discriminatory power of CAR and hsCRP for the risk of all-cause and cardiac death was observed. Whereas, a significant increase in the discriminatory power of ALB for the risk of death was observed. The HR (95%CI) for all-cause death in the high ALB group was 0.43 (0.29–0.63) (P < 0.001), and the HR (95%CI) for cardiac death was 0.31 (0.17–0.58) (P < 0.001) (S4 Table).

### Subgroup analysis

To further investigate the association of CAR with all-cause and cardiac death, stratified analyses by gender, age, eGFR, diabetes and COPD were performed (Table 3). Among participants of different gender, age and eGFR, or with and without diabetes and COPD, there was no significant difference in the association of CAR with all-cause or cardiac death (P for interaction > 0.05). When using restricted cubic spline plots to depict CAR as a continuous variable, a linear correlation was observed between CAR and both cardiac and all-cause death rates in subgroups stratified by gender, age, eGFR, diabetes and COPD (Fig 4, S1 Fig).

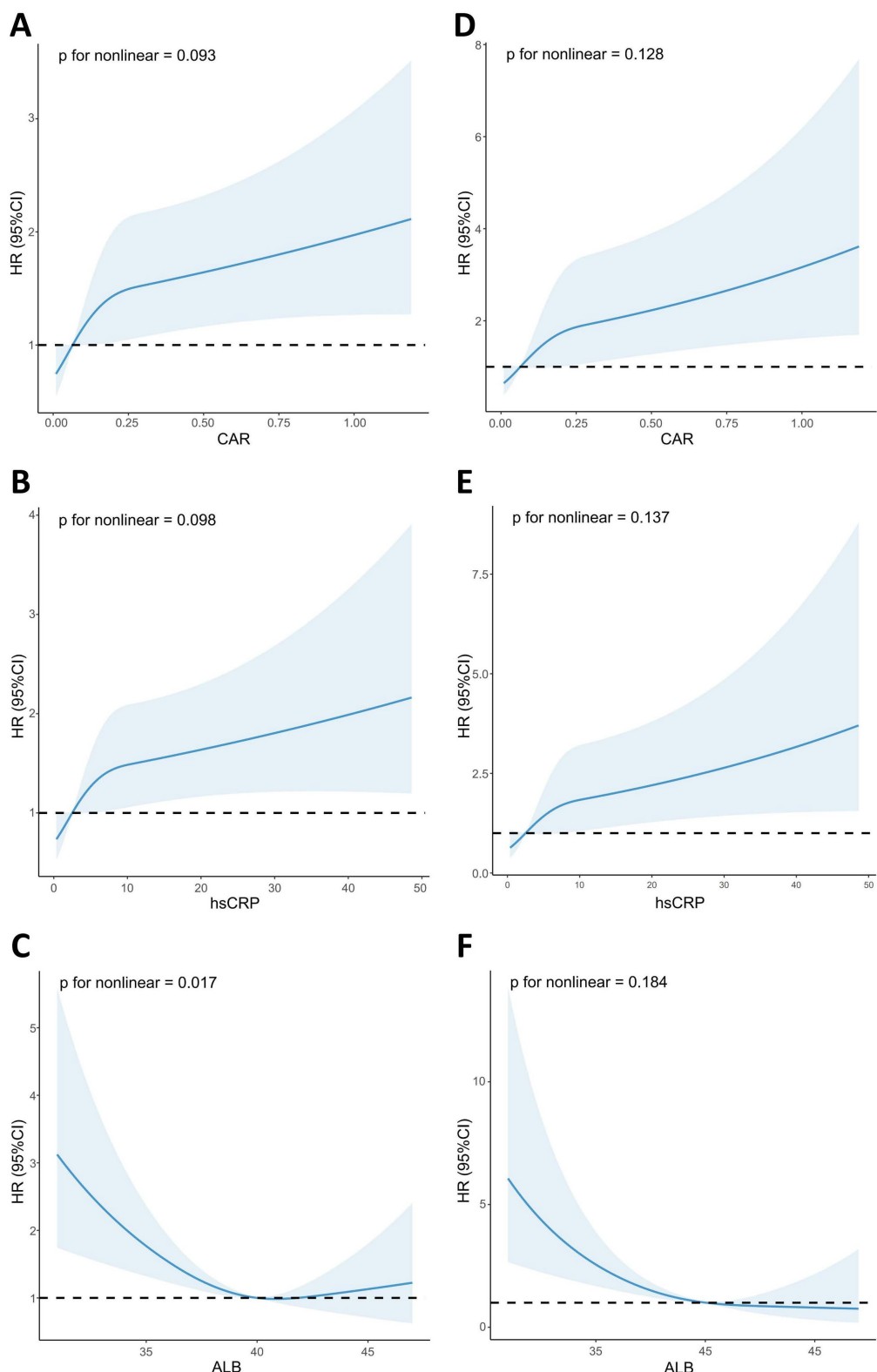

**Fig 2. Restricted spline curves for the associations between CAR, hsCRP, ALB and all-cause death (A–C) and cardiovascular death (D–F) in cardiovascular disease patients.**

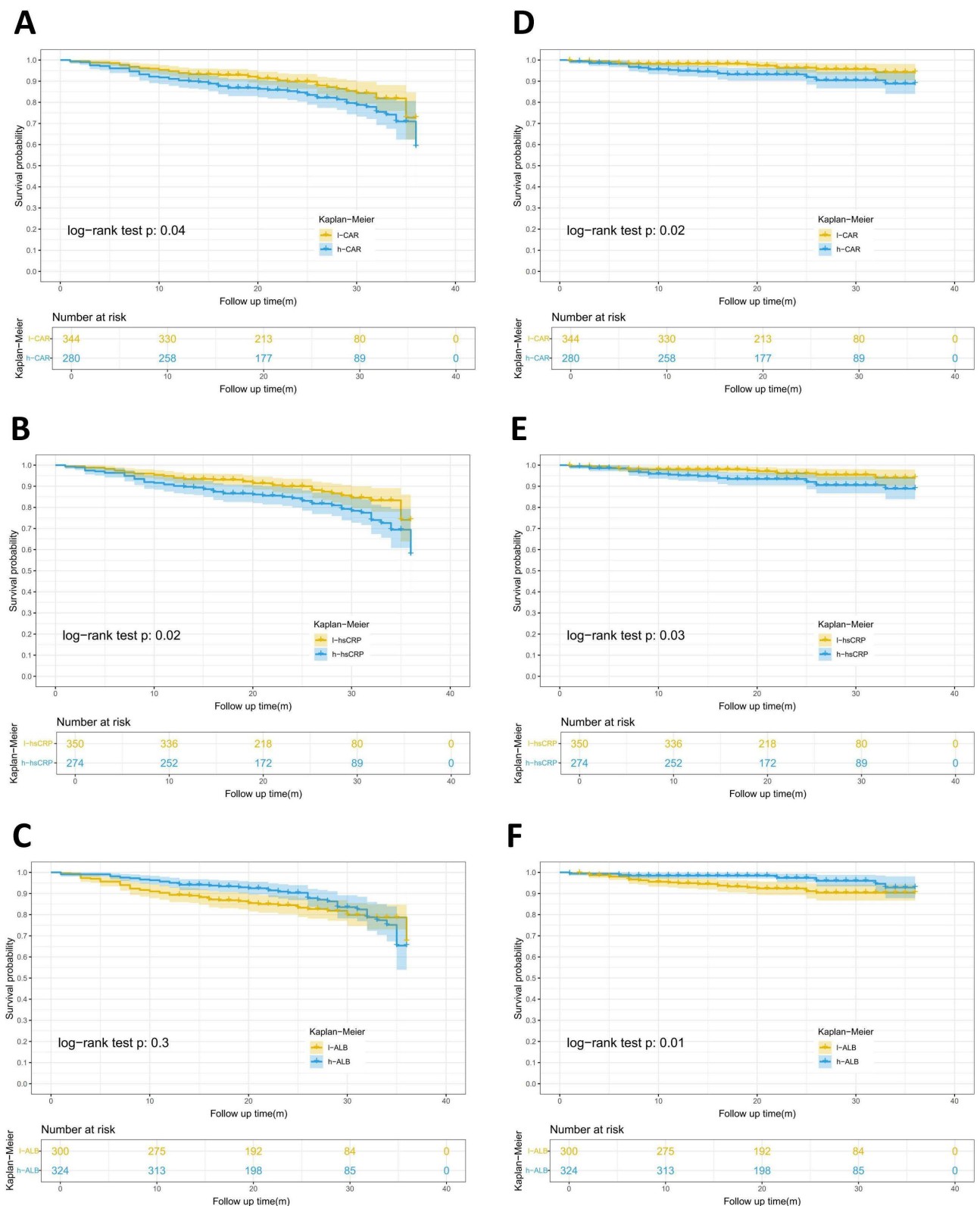

**Fig 3. Kaplan-Meier survival curves for all-cause death (A–C) and cardiac death (D–F) associated with CAR, hsCRP and ALB.**

**Table 2. Univariate and multivariable Cox regression analysis for predictors of cardiac death.**

| Variables | HR (95%CI) | P value |
|---|---|---|
| **Univariate analysis** | | |
| Age | 1.09 (1.04-1.13) | < 0.001 |
| Female | 0.35 (0.15-0.80) | 0.013 |
| Body mass index | 0.98 (0.93-1.03) | 0.484 |
| Physical exercise | 0.95 (0.49-1.85) | 0.878 |
| Diabetes | 2.60 (1.32-5.11) | 0.006 |
| Anemia | 0.90 (0.46-1.76) | 0.757 |
| Asthma | 1.13 (0.44-2.90) | 0.806 |
| Depression | 1.75 (0.89-3.44) | 0.106 |
| Segmented neutrophils | 1.12 (0.94-1.32) | 0.201 |
| Platelet | 1.00 (1.00-1.01) | 0.652 |
| Plasma glucose | 1.00 (0.99-1.01) | 0.760 |
| Total cholesterol | 0.99 (0.98-1.00) | 0.077 |
| Triglycerides | 1.00 (1.00-1.01) | 0.757 |
| LDL-C | 0.99 (0.98-1.01) | 0.263 |
| HDL-C | 0.99 (0.96-1.01) | 0.343 |
| AST | 1.02 (0.99-1.05) | 0.200 |
| eGFR | 0.98 (0.97-1.00) | 0.007 |
| Hypertension | 1.16 (0.53-2.55) | 0.715 |
| COPD | 2.02 (1.03-3.98) | 0.041 |
| Cancer | 1.75 (0.84-3.65) | 0.133 |
| CAR | 2.31 (1.15-4.65) | 0.019 |
| hsCRP | 2.13 (1.07-4.23) | 0.031 |
| ALB | 0.42 (0.21-0.86) | 0.018 |
| **Multivariable analysis** | | |
| CAR | 2.99 (1.44-6.22) | 0.003 |
| hsCRP | 2.63 (1.28-5.43) | 0.009 |
| ALB | 0.41 (0.20-0.87) | 0.021 |

Multivariable analysis was adjusted for Female, Age, Diabetes, Depression, Total cholesterol, AST, eGFR, COPD and Cancer.

## Discussion

In this study utilizing the NHANES database, we discovered that elevated CAR, a marker of inflammation and nutrition, was associated with an increased risk of all-cause and cardiac death in community-dwelling CHD patients. Among the covariates, we found that gender, age, diabetes, depression, segmented neutrophils, AST, eGFR, hypertension, COPD and cancer were all significantly associated with all-cause death in univariate COX regression analysis. Moreover, the impact of diabetes, depression, hypertension, COPD and cancer on the risk of death was even greater than that of CAR. The reason is that all-cause death also includes deaths caused by these diseases, but not all of these diseases are related to inflammatory risk. Therefore, we believe that CAR is not accurate enough in predicting the risk of all-cause death. However, in terms of cardiac death, only gender, age, diabetes, eGFR and COPD were significantly associated with an increased risk of death in univariate COX analysis. In subgroup analysis, there was no significant difference in the association between CAR and the risk of death among subjects with different genders, ages, eGFR, history of diabetes and COPD. Moreover, the impact of these indicators on the risk of cardiac death was far lower than that of CAR, which

**Table 3. Subgroup analysis for the association between CAR and mortality.**

| | All-cause death (n, %) | HR (95%CI) | P value | P-int | Cardiac death (n, %) | HR (95%CI) | P value | P-int |
|---|---|---|---|---|---|---|---|---|
| **Gender** | | | | 0.822 | | | | 0.792 |
| Male | 69 (18.8) | 1.70 (1.02-2.84) | 0.043 | | 28 (7.6) | 2.77 (1.23-6.24) | 0.014 | |
| Female | 28 (10.9) | 2.42 (0.99-5.94) | 0.053 | | 7 (2.7) | 6.81 (0.84-54.85) | 0.072 | |
| **Age** | | | | 0.736 | | | | 0.865 |
| <65 years | 79 (22.5) | 2.43 (0.79-7.47) | 0.122 | | 30 (8.5) | 7.41 (0.23-234.47) | 0.256 | |
| ≥65 years | 18 (6.6) | 1.75 (1.07-2.85) | 0.025 | | 5 (1.8) | 2.96 (1.36-6.42) | 0.006 | |
| **eGFR** | | | | 0.839 | | | | 0.541 |
| <60mL/min/1.73m$^2$ | 37 (23.0) | 1.76 (0.80-3.89) | 0.162 | | 15 (9.3) | 2.02 (0.62-6.60) | 0.242 | |
| ≥60mL/min/1.73m$^2$ | 60 (13.0) | 1.76 (1.01-3.06) | 0.046 | | 20 (4.3) | 4.01 (1.52-10.59) | 0.005 | |
| **Diabetes** | | | | 0.580 | | | | 0.617 |
| Yes | 52 (22.2) | 1.38 (0.74-2.57) | 0.312 | | 21 (9.0) | 3.08 (1.12-8.41) | 0.029 | |
| No | 45 (11.5) | 2.11 (1.09-4.08) | 0.027 | | 14 (3.6) | 2.60 (0.85-7.91) | 0.093 | |
| **COPD** | | | | 0.166 | | | | 0.775 |
| Yes | 36 (22.8) | 3.29 (1.51-7.16) | 0.003 | | 14 (9.0) | 3.98 (1.14-13.85) | 0.030 | |
| No | 61 (13.0) | 1.38 (0.79-2.41) | 0.252 | | 21 (4.5) | 2.74 (1.05-7.10) | 0.039 | |

HR: Hazard ratio, CI: Confidence interval, P-int: p for interaction. Cardiac death was adjusted for gender, age, eGFR, diabetes, COPD, hypertension, AST, cancer and CAR. All-cause death was adjusted for gender, age, eGFR, diabetes, COPD and CAR.

not only demonstrated the correlation between cardiac death and inflammatory risk, but also ensured the more effective role of CAR in predicting the risk of cardiac death in CHD population. Additionally, we established a specific CAR value of 0.075 mg/g rather than a cut-off value to differentiate patients at high risk of inflammation from those at low risk, with high inflammation being linked to a higher risk of death. Furthermore, CAR exhibited superior discriminatory ability for the risk of cardiac death in community-dwelling CHD patients compared to hsCRP and ALB.

There is mounting evidence indicating that inflammatory markers based on blood cell and serological indicators such as systemic immune-inflammation index (SII) [22,23], systemic inflammatory response index (SIRI) [24,25], CAR [26–28] and C-reactive protein-albumin-lymphocyte index (CALLY) [29] are significantly correlated with the risk of cardiovascular disease-related mortality. However, hsCRP remains the most commonly used indicator in clinical practice. The discriminatory ability of other indicators for adverse cardiovascular events largely depends on multiple classifications within specific populations and lacks clear standard values for distinguishing levels of inflammation. For instance, in studies conducted in hospital settings predicting major adverse cardiovascular events (MACE) among 659 ACS patients found that hsCRP/prealbumin (PAB) was an independent predictor of MACE using median dichotomization for hsCRP/PAB [30]. Another study involving 664 percutaneous coronary intervention (PCI) patients divided them into three groups based on tertiles of CAR, it found that high CAR increased the risks associated with all-cause death, cardiac death and MACE [31]. A Chinese community-based research involving 62,067 participants revealed that high levels of CAR can increase the risk of CVD. When screening high-risk populations for CVD, we should pay special attention to those with a simultaneous increase in CRP and a decrease in albumin [32].

Several large population studies have recommended the use of 1 and 3 mg/L as the thresholds of hsCRP to distinguish patients with low, moderate and high risk of cardiovascular events [33,34]. It is worth noting that the cut-off point of hsCRP in patients with ACS may differ from that in asymptomatic patients. For example, a level of 3 mg/L is more useful in patients with stable coronary artery disease, while a level of 10 mg/L may have better predictive value in ACS [35]. However, > 10 mg/L of hsCRP may indicate the presence of other inflammatory diseases. The population included in this study was community CHD population, which was closer to stable CHD according to symptoms, therefore, >3 mg/L was used to define high hsCRP. The diagnostic criteria for serum albumin will vary due to different diagnostic instruments. However, for

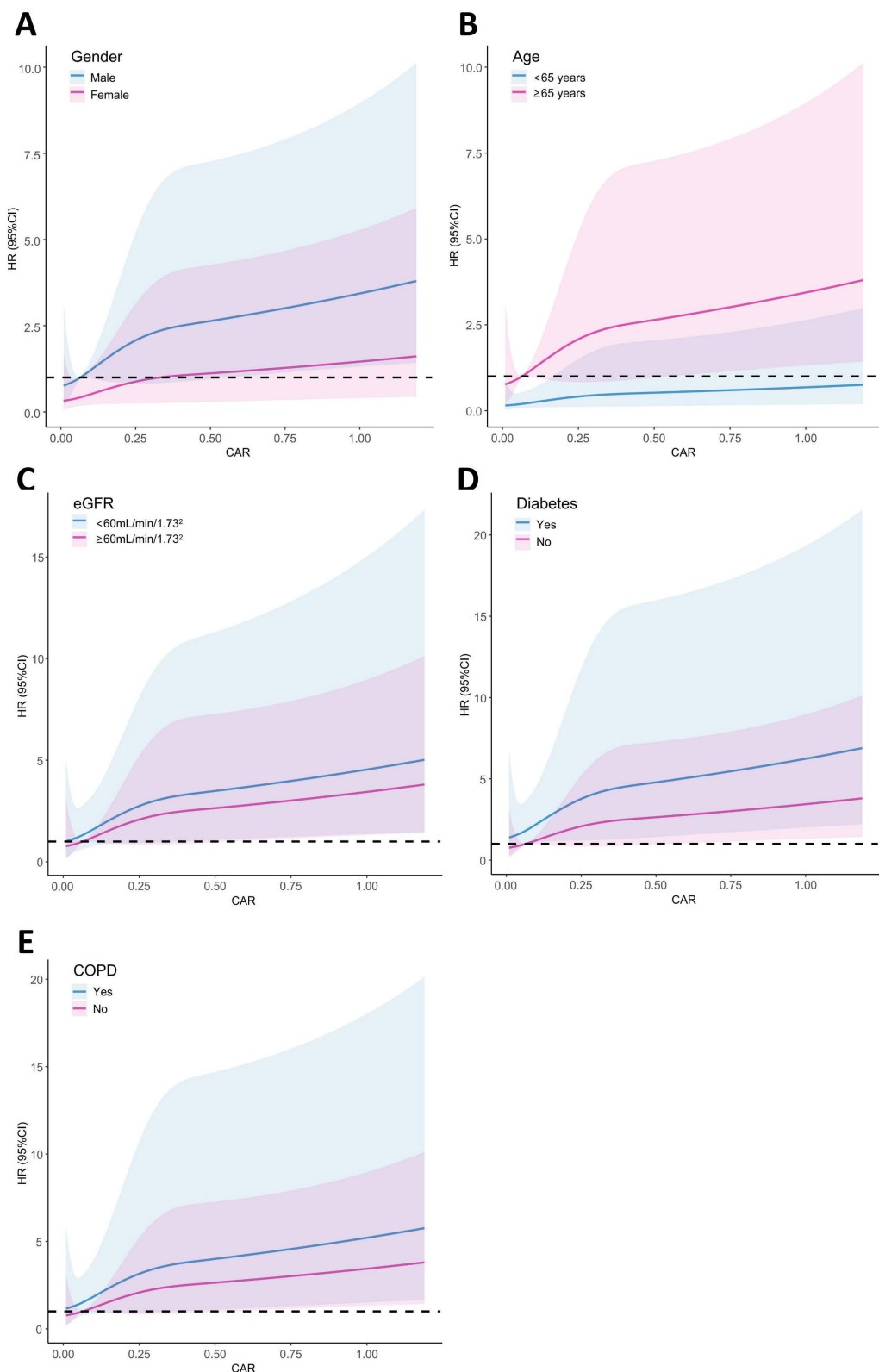

**Fig 4. Subgroup analysis of restricted cubic spline plots for the association between CAR and cardiac death by gender (A), age (B), eGFR (C), diabetes (D) and COPD (E).**

normal adults, albumin ≥40 g/L is considered normal [36]. Notwithstanding that the RCS curve has indicated that each unit increase in CAR is associated with a 78% increase in all-cause death risk and a 164% increase in cardiac death risk, it is not applicable for clinical analysis. Therefore, we used 3 mg/L divided by 40 g/L equals 0.075 mg/g as the cut-off value of CAR to distinguish between high inflammatory risk and low inflammatory risk. The study included a population from 2015 to 2018, with follow-up until December 31, 2019. In addition, hsCRP is an acute phase protein, mainly reflecting short-term disease status. To mitigate the bias caused by varying follow-up times, the study focused on a CHD population with a follow-up time of ≤ 36 months. After adjusting for other risk factors related to death, high CAR was associated with an 77% increased risk of all-cause death and a 199% increased risk of cardiac death. In terms of cardiac death risk discrimination, CAR outperformed hsCRP and ALB. When the follow-up time was extended to 60 months, the mortality risk of the high CAR and hsCRP groups was not as high as that of the low-value group. However, the discriminative ability of ALB for death risk significantly increased over this longer period. Although hsCRP can be used for CVD screening and risk stratification in short-term events due to its acute phase nature reflecting short-term disease status, it may not be as suitable for predicting long-term outcomes due to its large fluctuation range and lack of continuous measurement data in this study. On the other hand, ALB, being more stable over time as it is the main protein in plasma content less affected by follow-up duration compared to hsCRP or CAR, showed improved discriminative ability for long-term death risks as more subjects were followed up. Furthermore, although a U-shaped correlation between ALB and the 3-year all-cause death risk in patients with CHD was observed, when the risk was calculated in segments using the inflection point of 40.88, it was found that on the left side of the inflection point, ALB was linearly and negatively correlated with the increase in all-cause death risk, while on the right side of the inflection point, the increase in ALB had no statistical significance in relation to the increase in death risk. Nevertheless, in clinical practice, while assessing the death risk of patients through CAR, we still need to pay attention to whether ALB is much higher than the normal value due to hemoconcentration or immune system diseases in patients, as this can cause bias in the judgment of outcome events. Our findings are basically consistent with previous studies.

This study presents several innovative aspects. Firstly, it includes an investigation into the relationship between CAR and death risk in a community CHD population, which has not been previously reported in the literature. Secondly, the study determined an effective CAR cutoff value similar to hsCRP for distinguishing high- and low-inflammatory risk populations for death risk assessment. Thirdly, it found that CAR and hsCRP were more suitable for short-term death risk discrimination, while ALB was more suitable for long-term death risk discrimination. However, this study also has certain limitations. The sample size of the population studied was limited, and the diagnosis of CHD relied on a questionnaire survey rather than a large sample with accurately diagnosed individuals. Additionally, although a relatively effective CAR cut-off value was determined to distinguish inflammatory risk, the cut-off values of hsCRP and ALB need to be analyzed according to specific study populations due to differences in underlying diseases, regions and detection instruments. Besides, despite adjusting for potential risk factors of all-cause and cardiac death, this observational study cannot exclude the possibility of ignored or unmeasurable confounding factors. Finally, a portion of the population was included based on the symptoms of angina. Due to variations in disease severity, geographical and economic environments, it is unknown whether all or a suitable proportion of this population has the opportunity or is suitable for MRA/CTA examinations although they are conventional examinations for diagnosing CAD. Therefore, it is challenging to combine CAR and imaging examinations for risk prediction. However, with the increasing popularity of imaging examinations, it will be feasible to incorporate CAR and imaging examinations into risk stratification for prognosis-related research in the future.

## Conclusions

A CAR of 0.075 mg/g can effectively distinguish between high and low inflammation risk. A higher CAR is significantly associated with an increased risk of all-cause and cardiac death in the community-based CHD population.

## Supporting information

**S1 Fig.  Subgroup analysis of restricted cubic spline plots for the association between CAR and all-cause death by gender (A) age (B) eGFR (C) diabetes (D) and COPD (E).**
(JPG)

**S1 Table.  Baseline characteristics of cardiovascular disease patients according to different hsCRP groups.**
(DOCX)

**S2 Table.  Baseline characteristics of cardiovascular disease patients according to different ALB groups.**
(DOCX)

**S3 Table.  Univariate and multivariable Cox regression analysis for predictors of all-cause death.** Multivariable analysis was adjusted for female age AST eGFR hypertension diabetes COPD and cancer.
(DOCX)

**S4 Table.  Multivariable Cox regression analysis for predictors of all-cause and cardiac death at different follow-up times.**
(DOCX)

**S1 File.  Supplementary methods.**
(DOCX)

## Acknowledgments

We acknowledge the NHANES project members for their work in patient follow-up and data collection. We acknowledge the patients and their families for participating in this study.

## Author contributions

**Conceptualization:** Shangxun Zhou, Miaohan Qiu.

**Funding acquisition:** Yaling Han.

**Methodology:** Kexin Wang.

**Project administration:** Yaling Han.

**Supervision:** Jing Li, Yi Li, Yaling Han.

**Validation:** Shangxun Zhou, Miaohan Qiu, Kexin Wang.

**Visualization:** Ying Xu, Xuefei Mu.

**Writing – original draft:** Shangxun Zhou.

**Writing – review & editing:** Shangxun Zhou, Yixuan Duan, Daoshen Liu.

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
