## [Decision Letter · Decision Letter 0]

22 Oct 2024

PONE-D-24-23633Association of high-sensitivity C-reactive protein to albumin ratio with all-cause and cardiac death in coronary heart disease individuals: a retrospective NHANES studyPLOS ONE

Dear Dr. Han,

Thank you for submitting your manuscript to PLOS ONE. After careful consideration, we feel that it has merit but does not fully meet PLOS ONE’s publication criteria as it currently stands. Therefore, we invite you to submit a revised version of the manuscript that addresses the points raised during the review process.

We look forward to receiving your revised manuscript.

Kind regards,

Qian Wu

Academic Editor

PLOS ONE

Additional Editor Comments (if provided):

Reviewers' comments:

Reviewer's Responses to Questions

**Comments to the Author**

1. Is the manuscript technically sound, and do the data support the conclusions?

Reviewer #1: Partly

Reviewer #2: Partly

Reviewer #3: Partly

2. Has the statistical analysis been performed appropriately and rigorously? 

Reviewer #1: Yes

Reviewer #2: Yes

Reviewer #3: No

3. Have the authors made all data underlying the findings in their manuscript fully available?

Reviewer #1: Yes

Reviewer #2: Yes

Reviewer #3: Yes

4. Is the manuscript presented in an intelligible fashion and written in standard English?

Reviewer #1: Yes

Reviewer #2: Yes

Reviewer #3: Yes

5. Review Comments to the Author

Reviewer #1: The paper describes that a higher CAR is significantly associated with an increased risk of all-cause and cardiac death in the community-based CHD population, but there are still several issues that need to be addressed.

1.Please provide a flowchart detailing the process of screening the population.

2.The authors categorized the CAR at 0.075 mg/g to stratify the levels of inflammation. Please demonstrate the optimal cutoff value from the ROC curve.

3.The study results do not seem to support the descriptions in the discussion section. Please revise the discussion section carefully.For example, "In terms of cardiac death risk discrimination, CAR outperformed hsCRP and ALB, and the prediction model containing CAR had better predictive ability for all-cause death than ALB."

4.There is an inconsistency between the text and the table, such as on line 185. Please correct this and check other parts for similar discrepancies.

5. The statements in lines 220 and 221 appear to be contradictory. Please clarify or resolve the inconsistency.

6.There are some spelling errors in the text. Please correct them accordingly.

Reviewer #2: The study was interesting, but I have following comments to improve the manuscript.

1. The study aimed to explore the association of high-sensitivity C-reactive protein to albumin ratio with all-cause and cardiac death in coronary heart disease individuals by using the data from NHANES (2015-2018); however, I am curious about the delay in publishing these findings until 2024.

2. It has been reported that geographic disparities in cardiovascular health are prevalent among American adults (doi: 10.1016/j.mayocp.2020.12.034). Did the authors consider this variability when selecting their samples?

3. MR and CT imaging have emerged as promising complementary techniques for the primary diagnosis of coronary artery disease (CAD) and the detection of coronary atherosclerosis. There is publication based on the relationship between C-Reactive Protein/Albumin Ratio and Radiological Findings in Patients with COVID-19 (DOI:10.16899/jcm.900886). The authors might think to explore the inclusion of this relationship in their study.

Reviewer #3: Dear Author, Thank you for conducting this study. However, The following points should be considered:

The main issue with the current study is the dispersion of its content and analyses. To examine the relationship between variables and outcomes, using limited but precise and detailed analyses is preferable to employing numerous unclear and incomplete approaches. The multitude of analyses and tables has resulted in many important points and explanations regarding the results being overlooked in both the tables and the text. Therefore, it is advisable to rewrite the manuscript while considering these points. However, the following notes are also significant:

1- Many of the variables mentioned in Table 1, which could be related to the study's outcomes, have been excluded in other analyses, leaving only a limited number included. What was the reason for this?

2- It is recommended that variables with a significance level below 0.2 or 0.18 in univariate analysis be included in the multivariate model (to control for confounders, maximize information, increase the statistical power of the study, etc.); however, this study only selected variables with a significance level below 0.05.

3- For the classification of variables (such as smoking, diabetes, etc.), it is essential to provide references for the basis of classification.

4- The exclusion of individuals with missing hsCRP data may lead to selection bias, potentially skewing the results if those excluded had different mortality risks compared to those included.

5- While 624 participants may seem adequate, it could be insufficient for subgroup analyses, particularly when examining interactions across multiple covariates. Small sample sizes in subgroups can lead to unreliable estimates.

6- The U-shaped correlation between ALB and all-cause death risk is intriguing but requires further exploration and explanation. The implications of this finding for clinical practice are not sufficiently discussed.

7- Although Kaplan-Meier curves can be drawn based on cumulative hazard, since Kaplan-Meier and Cox models are fundamentally recognized as survival curves, it is better for the graphs to be drawn based on cumulative survival.

8- What was the purpose of using restricted cubic splines? There is no clear explanation regarding their results, and the discussion section does not tangibly address the dose-response relationship.

6. PLOS authors have the option to publish the peer review history of their article (what does this mean? ). If published, this will include your full peer review and any attached files.

**Do you want your identity to be public for this peer review?** For information about this choice, including consent withdrawal, please see our Privacy Policy .

Reviewer #1: No

Reviewer #2: No

Reviewer #3: No

---

## [Author Response · Author response to Decision Letter 1]

17 Nov 2024

Reviewer 1

Q1: Please provide a flowchart detailing the process of screening the population.

A1: We are extremely grateful for your comments on the manuscript. The flowchart detailing the process of screening the population has been added to the manuscript as Fig 1 in the Materials and Methods Section.

Fig 1. The flowchart detailing the process of screening the population.

Q2: The authors categorized the CAR at 0.075 mg/g to stratify the levels of inflammation. Please demonstrate the optimal cutoff value from the ROC curve.

A2: Thank you for the comments you put forward, highlighting the deficiency of the description in the manuscript. The ROC curve for cardiac death was calculated using logistic regression. The maximum value of the Youden index was 0.524, at which the CAR value was 0.0773486, the sensitivity was 0.743, and the specificity was 0.781 (Table 1). This was the optimal cut-off value. However, considering that the value of 0.077 is not convenient for calculation. Moreover, 0.075 mg/g was obtained from hsCRP = 3 (mg/L)/ALB = 40 (g/L). Therefore, it is undoubtedly more appropriate to use CAR = 0.075 mg/g.

Table 1. The cut-off value of CAR calculated by the ROC curve .

CAR value Sensitivity 1-Specialty Youden Index

0.0773486 0.743 0.219 0.524

0.0765809 0.743 0.221 0.522

0.0755362 0.743 0.222 0.521

0.0751677 0.743 0.224 0.519

0.0747171 0.743 0.226 0.517

0.074019 0.743 0.228 0.515

0.0735054 0.743 0.229 0.514

0.0729381 0.743 0.231 0.512

0.0667209 0.771 0.261 0.510

0.0725357 0.743 0.233 0.510

 ... ... ... ...

Q3: The study results do not seem to support the descriptions in the discussion section. Please revise the discussion section carefully.For example, "In terms of cardiac death risk discrimination, CAR outperformed hsCRP and ALB, and the prediction model containing CAR had better predictive ability for all-cause death than ALB."

A3: Thank you for your comment. For the sake of the concentration of the article content, we have deleted the relevant content of ROC curves, because this part makes a limited contribution to the main results of the manuscript. The statements in the Discussion Section have been modified.

“...In terms of cardiac death risk discrimination, CAR outperformed hsCRP and ALB...”

Q4: There is an inconsistency between the text and the table, such as on line 185. Please correct this and check other parts for similar discrepancies..

A4: Thank you for your comments. We carefully reviewed the results of the manuscript once again. When the subgroup analysis indicated that there was no interaction between CAR and each covariate, it was meaningless to analyze the differences in the death risk among the subgroups. Therefore, the part regarding the subgroup analysis of diabetic patients was deleted. The remaining table descriptions have also been meticulously cross-checked.

Q5: The statements in lines 220 and 221 appear to be contradictory. Please clarify or resolve the inconsistency.

A5: Thank you for your valuable comments. In some studies, CRP/ALB is called CAR, and in others, it is hsCRP/PAB. However, in our study, it is hsCRP/ALB. The cause lies in the fact that the patient information collected in each study varies, but the obtained conclusions are basically similar. To eliminate any ambiguity in expression, we have modified the Discussion Section of the manuscript.

“...that hsCRP/prealbumin (PAB) was an independent predictor of MACE using median dichotomization for hsCRP/PAB...”

Q6: There are some spelling errors in the text. Please correct them accordingly.

A6: Thank you for your meticulous examination. This error has already been rectified in the manuscript.

Reviewer 2

Q1: The study aimed to explore the association of high-sensitivity C-reactive protein to albumin ratio with all-cause and cardiac death in coronary heart disease individuals by using the data from NHANES (2015-2018); however, I am curious about the delay in publishing these findings until 2024.

A1: Thank you for your question. We equally express our regret for the problems you mentioned. The NHANES program suspended field operations in March 2020 due to the coronavirus disease 2019 (COVID-19) pandemic. As a result, data collection for the NHANES 2019-2020 cycle was not completed and the collected data are not nationally representative. Consequently, the data of the years 2019 - 2020 are beyond the consideration of this research.

Although the NHANES database was reactivated in August 2021, the NDI data associated with the NCHS was not entirely accessible. The follow-up data related to deaths was only updated until December 31, 2019. Thus, the data after 2021, although containing various covariates including CAR, lacked outcome events and were therefore excluded. Nevertheless, we anticipate that if there are more updates to the mortality data related to the NCHS, we will incorporate more studies with larger sample sizes to reinforce the conclusions of this research.

Q2: It has been reported that geographic disparities in cardiovascular health are prevalent among American adults (doi: 10.1016/j.mayocp.2020.12.034). Did the authors consider this variability when selecting their samples?

A2: Thank you very much for your valuable question. NHANES uses a stratified multistage sampling design to obtain a representative sample of US residents. The sampling plan consists of four stages: counties, segments, households and individuals. Moreover, the populations of different races have corresponding distributions in the groups. Therefore, the data we employed are representative to a certain extent. Nevertheless, given the small sample size of this study and the selection of the CHD population as the research subjects, a certain degree of bias may occur. The restriction of a slightly smaller sample size has been incorporated into the Discussion Section.

“...The sample size of the population studied was limited, and the diagnosis of CHD relied on a questionnaire survey rather than a large sample with accurately diagnosed individuals...”

Q3: MR and CT imaging have emerged as promising complementary techniques for the primary diagnosis of coronary artery disease (CAD) and the detection of coronary atherosclerosis. There is publication based on the relationship between C-Reactive Protein/Albumin Ratio and Radiological Findings in Patients with COVID-19 (DOI:10.16899/jcm.900886). The authors might think to explore the inclusion of this relationship in their study.

A3: Thank you for your comments and valuable suggestions. The article (DOI:10.16899/jcm.900886) aimed to determine the C-reactive protein/Albumin ratio (CAR) values of computed tomography (CT) -positive COVID-19 patients and CT-negative COVID-19 patients and to investigate the relationship between C-reactive protein/Albumin ratio and radiological images of patients. In cases where CT is contraindicated (such as pregnancy), CAR can be used to indicate lung involvement or to follow-up patients with pulmonary involvement.

However, there appears to be a disparity in the case of cardiovascular diseases. The imaging of cardiovascular diseases is the conventional examination for diagnosing CAD. Currently, an increasing number of studies are based on imaging examinations such as OCT, FFR and CT-FFR to make treatment decision choices. If the disease progresses to this stage, the value of imaging examinations is far superior to that of CAR. CAR can merely serve as a reference for determining the risk of adverse cardiovascular events with poor long-term prognosis in patients. Different from respiratory diseases, there are multiple diagnostic methods for CHD. Even if radiological examinations are not feasible, patients can still undergo non-invasive and non-radiation examinations like electrocardiography, echocardiography and hematological tests (such as cTNT and CK-MB) to determine the severity of CAD. Finally, although the risk of cardiac death in the CHD population with high CAR in this study was significantly higher than that in the low CAR group, there are considerable limitations in population selection. This study is only applicable to the community-based CHD population, and a portion was included based on the symptoms of angina. Due to variations in geographical and economic environments, it is unknown whether all or a suitable proportion of this population has the opportunity or is suitable for MRA/CTA examinations. Therefore, it is challenging to combine CAR and imaging examinations for risk prediction.

Nevertheless, an increasing number of institutions, including our research center, have initiated studies related to imaging in patients with CHD. Subsequently, the imaging indicators and CAR values of this in-hospital patient population can be directly utilized for research on the risk of adverse cardiovascular events. This will be an interesting research outcome and possess better clinical applicability. The relevant content has been discussed in the Discussion Section of the manuscript:

“...Finally, a portion of the population was included based on the symptoms of angina. Due to variations in disease severity, geographical and economic environments, it is unknown whether all or a suitable proportion of this population has the opportunity or is suitable for MRA/CTA examinations although they are conventional examinations for diagnosing CAD. Therefore, it is challenging to combine CAR and imaging examinations for risk prediction. However, with the increasing popularity of imaging examinations, it will be feasible to incorporate CAR and imaging examinations into risk stratification for prognosis-related research in the future....”

Reviewer 3

The main issue with the current study is the dispersion of its content and analyses. To examine the relationship between variables and outcomes, using limited but precise and detailed analyses is preferable to employing numerous unclear and incomplete approaches. The multitude of analyses and tables has resulted in many important points and explanations regarding the results being overlooked in both the tables and the text. Therefore, it is advisable to rewrite the manuscript while considering these points.

A: Thank you for your meticulous review and suggestions for this paper. After scrupulously contemplating the integrity of the article and the interpretability of the content, we re-wrote the manuscript and eliminated a portion of the content that was not highly relevant to the main results.

Q1: Many of the variables mentioned in Table 1, which could be related to the study's outcomes, have been excluded in other analyses, leaving only a limited number included. What was the reason for this?

A1: Thank you for the question. Although some data, such as pulse, physical exercise and anemia, exhibited differences between the high CAR and low CAR groups in the baseline table, a univariate COX regression analysis revealed that they had no correlation with the primary outcome event of all-cause death. Hence, these indicators were excluded from the multivariate COX regression. However, in accordance with your suggestion in Question 2, all variables with a significance level lower than 0.2 in the univariate analysis were incorporated into the multivariate model to control confounding factors, maximize information and increase the statistical power of the study. The specific univariate and multivariate Cox regression tables are presented uniformly in Question 2, and the relevant statistical data in the manuscript has been rewritten.

Q2: It is recommended that variables with a significance level below 0.2 or 0.18 in univariate analysis be included in the multivariate model (to control for confounders, maximize information, increase the statistical power of the study, etc.); however, this study only selected variables with a significance level below 0.05.

A2: Thank you for the comments you put forward, highlighting the deficiency of the manuscript. In accordance with your suggestion, we have re-included all the variables with a significant difference of less than 0.2 in the univariate analysis into the multivariate Cox regression analysis and have rewritten the paper. Among them, there were an excessive number of missing values for LDL-C. Therefore, it was excluded from the multivariate Cox regression analysis. The results of the univariate and multivariate analyses related to all-cause and cardiac death are as follows:

Table 1. Univariate and multivariable Cox regression analysis for predictors of all-cause death.

Variables Univariate analysis Multivariable analysis

HR (95%CI) P value HR (95%CI) P value

Age 1.07 (1.05-1.10) < 0.001 1.07 (1.04-1.10) < 0.001

Female 0.56 (0.36-0.86) 0.009 0.60 (0.37-0.98) 0.043

Body mass index 0.98 (0.95-1.01) 0.174 0.96 (0.92-0.99) 0.034

Physical exercise 0.95 (0.64-1.41) 0.794 - -

Diabetes 2.03 (1.36-3.03) 0.001 1.75 (1.73-2.71) 0.012

Anemia 0.94 (0.63-1.41) 0.766 - -

Asthma 1.64 (0.99-2.70) 0.055 1.18 (0.68-2.07) 0.555

Depression 1.94 (1.29-2.90) 0.001 2.19 (1.42-3.36) < 0.001

Segmented neutrophils 1.21 (1.10-1.33) < 0.001 1.13 (1.02-1.25) 0.026

Platelet 1.00 (0.99-1.00) 0.092 1.00 (1.00-1.01) 0.830

Plasma glucose 1.00 (1.00-1.01) 0.760 - -

Total cholesterol 1.00 (0.99-1.00) 0.090 1.00 (0.99-1.00) 0.730

Triglycerides 1.00 (1.00-1.01) 0.815 - -

LDL-C 0.76 (0.54-1.08) 0.128 - -

HDL-C 1.00 (0.98-1.01) 0.579 - -

AST 1.02 (1.10-1.04) 0.046 1.02 (1.00-1.04) 0.048

eGFR 0.99 (0.98-0.99) < 0.001 1.00 (0.99-1.01) 0.656

Hypertension 1.78 (1.16-2.73) 0.009 1.13 (0.72-1.80) 0.592

COPD 1.84 (1.22-2.78) 0.004 1.39 (0.89-2.20) 0.152

Cancer 2.46 (1.63-3.73) < 0.001 1.34 (0.86-2.08) 0.201

CAR 1.51 (1.01-2.25) 0.045 1.77 (1.15-2.74) 0.010

hsCRP 1.63 (1.09-2.44) 0.018 1.88 (1.22-2.90) 0.004

ALB 0.80 (0.54-1.19) 0.268 0.71 (0.46-1.08) 0.107

Multivariable analysis was adjusted for Age, Female, Body mass index, Diabetes, Asthma, Depression, Segmented neutrophils, Platelet, Total cholesterol, AST, eGFR, Hypertension, COPD and Cancer.

Abbreviations: high-sensitivity C-reactive protein to albumin ratio, CAR; chronic obstructive pulmonary disease, COPD; low-density lipoprotein cholesterol, LDL-C; high-density lipoprotein cholesterol, HDL-C; high-sensitivity C-reactive protein, hsCRP; albumin, ALB; Aspartate Aminotransferase, AST; estimated glomerular filtration rate, eGFR.

Table 2. Univariate and multivariable Cox regression analysis for predictors of cardiac death.

Variables Univariate analysis Multivariable analysis

HR (95%CI) P value HR (95%CI) P value

Age 1.09 (1.04-1.13) < 0.001 1.10 (1.04-1.15) < 0.001

Female 0.35 (0.15-0.80) 0.013 0.39 (0.17-0.94) 0.036

Body mass index 0.98 (0.93-1.03) 0.484 - -

Physical exercise 0.95 (0.49-1.85) 0.878 - -

Diabetes 2.60 (1.32-5.11) 0.006 1.90 (0.95-3.80) 0.068

Anemia 0.90 (0.46-1.76) 0.757 - -

Asthma 1.13 (0.44-2.90) 0.806 - -

Depression 1.75 (0.89-3.44) 0.106 2.01 (0.99-4.05) 0.052

Segmented neutrophils 1.12 (0.94-1.32) 0.201 - -

Platelet 1.00 (1.00-1.01) 0.652 - -

Plasma glucose 1.00 (0.99-1.01) 0.760 - -

Total cholesterol 0.99 (0.98-1.00) 0.077 1.00 (0.99-1.01) 0.404

Triglycerides 1.00 (1.00-1.01) 0.757 - -

LDL-C 0.99 (0.98-1.01) 0.263 - -

HDL-C 0.99 (0.96-1.01) 0.343 - -

AST 1.02 (0.99-1.05) 0.200 1.03 (1.00-1.06) 0.096

eGFR 0.98 (0.97-1.00) 0.007 1.00 (0.98-1.01) 0.581

Hypertension 1.16 (0.53-2.55) 0.715 - -

COPD 2.02 (1.03-3.98) 0.041 1.55 (0.78-3.09) 0.214

Cancer 1.75 (0.84-3.65) 0.133 0.93 (0.43-2.02) 0.851

CAR 2.31 (1.15-4.65) 0.019 2.99 (1.44-6.22) 0.003

hsCRP 2.13 (1.07-4.23) 0.031 2.63 (1.28-5.43) 0.009

ALB 0.42 (0.21-0.86) 0.018 0.41 (0.20-0.87) 0.021

Multivariable analysis was adjusted for Female, Age, Diabetes, Depression, Total cholesterol, AST, eGFR, COPD and Cancer.

Q3: For the classification of variables (such as smoking, diabetes, etc.), it is essential to provide references for the basis of classification.

A3: Thank you for your suggestion. The relevant cited literature has been supplemented in the Supplementary Methods.

Q4: The exclusion of individuals with missing hsCRP data may lead to selection bias, potentially skewing the results if those excluded had different mortality risks compared to those included.

A4: Thank you for the comments. In

---

## [Decision Letter · Decision Letter 1]

19 Jan 2025

PONE-D-24-23633R1Association of high-sensitivity C-reactive protein to albumin ratio with all-cause and cardiac death in coronary heart disease individuals: a retrospective NHANES studyPLOS ONE

Dear Dr. Han,

Thank you for submitting your manuscript to PLOS ONE. After careful consideration, we feel that it has merit but does not fully meet PLOS ONE’s publication criteria as it currently stands. Therefore, we invite you to submit a revised version of the manuscript that addresses the points raised during the review process.

Please make peer-to-peer modifications to the reviewer's comments.

We look forward to receiving your revised manuscript.

Kind regards,

Qian Wu

Academic Editor

PLOS ONE

Journal Requirements:

Reviewers' comments:

Reviewer's Responses to Questions

**Comments to the Author**

1. If the authors have adequately addressed your comments raised in a previous round of review and you feel that this manuscript is now acceptable for publication, you may indicate that here to bypass the “Comments to the Author” section, enter your conflict of interest statement in the “Confidential to Editor” section, and submit your "Accept" recommendation.

Reviewer #2: All comments have been addressed

Reviewer #4: (No Response)

2. Is the manuscript technically sound, and do the data support the conclusions?

Reviewer #2: (No Response)

Reviewer #4: Yes

3. Has the statistical analysis been performed appropriately and rigorously? 

Reviewer #2: (No Response)

Reviewer #4: Yes

4. Have the authors made all data underlying the findings in their manuscript fully available?

Reviewer #2: (No Response)

Reviewer #4: Yes

5. Is the manuscript presented in an intelligible fashion and written in standard English?

Reviewer #2: (No Response)

Reviewer #4: No

6. Review Comments to the Author

Reviewer #2: (No Response)

Reviewer #4: Introduction

There are too many references grouped together, such as in lines 39 and 45 (e.g., [1-5], [9-13]). This can weaken the important studies that should be highlighted, and may include incorrect citations, for example, citation 14 refers to COVID-19, not CVD, and citation 16 is a review that is not very relevant to the statements made. The authors need to correct these errors and any similar issues.

Results

In the materials and methods section, two outcomes are mentioned: all-time cause of death and cardiac death (line 77). However, the results section also includes cancer death as an outcome. If cancer death is not part of all-cause death, the authors should define all-cause death and clarify the role of cancer in the overall analysis.

Lines 222-224 contain an incomplete thought.

Reference 23 is not relevant to the statement provided.

Discussion

While the results are presented clearly, the discussion does not address the results effectively. This section could be enhanced by considering the potential effects of various covariates and the clinical and demographic characteristics of the study population on inflammation risk and CAR outcomes.

References

Authors need to ensure that references are cited correctly

7. PLOS authors have the option to publish the peer review history of their article (what does this mean? ). If published, this will include your full peer review and any attached files.

**Do you want your identity to be public for this peer review?** For information about this choice, including consent withdrawal, please see our Privacy Policy .

Reviewer #2: No

Reviewer #4: No

---

## [Author Response · Author response to Decision Letter 2]

2 Feb 2025

Reviewer 4

Q1: Introduction

There are too many references grouped together, such as in lines 39 and 45 (e.g., [1-5], [9-13]). This can weaken the important studies that should be highlighted, and may include incorrect citations, for example, citation 14 refers to COVID-19, not CVD, and citation 16 is a review that is not very relevant to the statements made. The authors need to correct these errors and any similar issues.

A1: Thank you for your suggestion. We have meticulously scrutinized each of the cited references to ensure their compliance with the citation standards. In citation 14, although CAR and other inflammatory indicators were mentioned, they were indeed irrelevant to CVD and have thus been deleted. Nevertheless, in citation 16, it was repeatedly stated that multiple studies have demonstrated that hsCRP is associated with the residual inflammatory risk of ASCVD and capable of guiding the treatment. The relevant statements are as follows, and we contend that they are pertinent to the description in this article.

The description in citation 16:

“...based on the magnitude of risk associations and availability of clinical assays, hsCRP emerged as the benchmark biomarker for inflammatory risk in ASCVD...”

“...A large body of evidence supports the use of hsCRP in primary prevention to guide therapeutic decision-making...”

“Although not yet part of routine clinical practice, hsCRP is also informative in secondary prevention. Most studies have observed positive associations between hsCRP and CV events...”

Q2: Results

In the materials and methods section, two outcomes are mentioned: all-time cause of death and cardiac death (line 77). However, the results section also includes cancer death as an outcome. If cancer death is not part of all-cause death, the authors should define all-cause death and clarify the role of cancer in the overall analysis.

A2: Thank you for the question. In the death events defined by NDI, the causes of death encompassed diseases of heart, malignant neoplasms, chronic lower respiratory diseases, accidents, cerebrovascular diseases, alzheimer’s disease, diabetes mellitus, influenza and pneumonia, nephropathy and other causes. Cancer death (malignant neoplasms) was a component of all-cause death. The definition of all-cause death was also appended to the Materials and Methods section, with the added content as follows. Cancer death was presented in the baseline data since cancer death was the second major cause of death in the study population after cardiac death, to observe whether there was any correlation between CAR and cancer death.

“...The causes of all-cause death encompasses diseases of heart, malignant neoplasms, chronic lower respiratory diseases, accidents, cerebrovascular diseases, alzheimer’s disease, diabetes mellitus, influenza and pneumonia, nephropathy and other causes...”

Q3: Lines 222-224 contain an incomplete thought..

A3: Thank you for your comments. The incomplete description has been rephrased as follows:

“...A Chinese community-based research involving 62,067 participants revealed that high levels of CAR can increase the risk of CVD. When screening high-risk populations for CVD, we should pay special attention to those with a simultaneous increase in CRP and a decrease in albumin...”

Q4: Reference 23 is not relevant to the statement provided.

A4: Thank you for the meticulous review. This citation is an incorrect citation and contains incomplete information, and has thereby been eliminated from the manuscript.

Q5: Discussion

While the results are presented clearly, the discussion does not address the results effectively. This section could be enhanced by considering the potential effects of various covariates and the clinical and demographic characteristics of the study population on inflammation risk and CAR outcomes.

A5: Thank you for the question, highlighting the deficiency of the manuscript. We have included the further elaboration of the results section within the Discussion, as follows:

“...Among the covariates, we found that gender, age, diabetes, depression, segmented neutrophils, AST, eGFR, hypertension, COPD and cancer were all significantly associated with all-cause death in univariate COX regression analysis. Moreover, the impact of diabetes, depression, hypertension, COPD and cancer on the risk of death was even greater than that of CAR. The reason is that all-cause death also includes deaths caused by these diseases, but not all of these diseases are related to inflammatory risk. Therefore, we believe that CAR is not accurate enough in predicting the risk of all-cause death. However, in terms of cardiac death, only gender, age, diabetes, eGFR and COPD were significantly associated with an increased risk of death in univariate COX analysis. In subgroup analysis, there was no significant difference in the association between CAR and the risk of death among subjects with different genders, ages, eGFR, history of diabetes and COPD. Moreover, the impact of these indicators on the risk of cardiac death was far lower than that of CAR, which not only demonstrated the correlation between cardiac death and inflammatory risk, but also ensured the more effective role of CAR in predicting the risk of cardiac death in CHD population...”

Q6: References

Authors need to ensure that references are cited correctly

A6: Thank you for your suggestion. Each reference citation has been meticulously examined to ensure its compliance with the citation requirements and its correct formatting.

---

## [Decision Letter · Decision Letter 2]

19 Mar 2025

Association of high-sensitivity C-reactive protein to albumin ratio with all-cause and cardiac death in coronary heart disease individuals: a retrospective NHANES study

PONE-D-24-23633R2

Dear Dr. Han,

We’re pleased to inform you that your manuscript has been judged scientifically suitable for publication and will be formally accepted for publication once it meets all outstanding technical requirements.

Kind regards,

Qian Wu

Academic Editor

PLOS ONE

Additional Editor Comments (optional):

Reviewers' comments:

Reviewer's Responses to Questions

**Comments to the Author**

1. If the authors have adequately addressed your comments raised in a previous round of review and you feel that this manuscript is now acceptable for publication, you may indicate that here to bypass the “Comments to the Author” section, enter your conflict of interest statement in the “Confidential to Editor” section, and submit your "Accept" recommendation.

Reviewer #4: All comments have been addressed

2. Is the manuscript technically sound, and do the data support the conclusions?

Reviewer #4: Yes

3. Has the statistical analysis been performed appropriately and rigorously? 

Reviewer #4: Yes

4. Have the authors made all data underlying the findings in their manuscript fully available?

Reviewer #4: Yes

5. Is the manuscript presented in an intelligible fashion and written in standard English?

Reviewer #4: Yes

6. Review Comments to the Author

Reviewer #4: The revisions have addressed my comments in a clear and comprehensive way, and I am satisfied with them.

7. PLOS authors have the option to publish the peer review history of their article (what does this mean? ). If published, this will include your full peer review and any attached files.

**Do you want your identity to be public for this peer review?** For information about this choice, including consent withdrawal, please see our Privacy Policy .

Reviewer #4: No

---

## [Editor Report · Acceptance letter]

PONE-D-24-23633R2

PLOS ONE

Dear Dr. Han,

I'm pleased to inform you that your manuscript has been deemed suitable for publication in PLOS ONE. Congratulations! Your manuscript is now being handed over to our production team.

Kind regards,

on behalf of

Dr. Qian Wu

Academic Editor

PLOS ONE